# RUBRICROBUSTNESS: Evaluating the Sensitivity of Rubrics-Based Benchmarks to Simple Perturbations

**Manasi Sharma** [1]

## Abstract

The advancement of Large Language Models (LLMs) into higher-level reasoning domains has rendered traditional heuristic evaluators insufficient for long-form open-ended responses, precipitating the widespread adoption of rubric-based benchmarks. While these frameworks utilize expert-curated criteria and LLM-as-a-judge to assess open-ended generation, the intrinsic robustness of these evaluation harnesses to fundamental validity assessments remains critically under-investigated. To bridge this gap, we introduce RUBRICROBUSTNESS, a systematic sensitivity analysis framework that subjects these benchmarks to three common sense perturbations: *semantic negation*, *stochastic deletion* and *irrelevant addition*. We investigate the extent to which manipulating the semantic veracity of a model's response impacts its resulting score by applying the robustness framework to two of the most popular rubrics-based benchmarks: HealthBench and WildBench. Our findings reveal systematic vulnerabilities: while both benchmarks respond sharply to semantic negation (e.g., degradation slopes of approximately $-0.38$ on HealthBench and $-0.55$ on WildBench), they are substantially less responsive to irrelevant addition, often requiring over 35% of sentences to be perturbed before inducing even a 25% score drop. We argue that perturbation-based sensitivity analyses of this form are a necessary prerequisite for validating rubric coverage, ensuring that automated evaluation frameworks reliably penalize basic semantic failures. We will release our framework as an open-source tool for building more resilient benchmarks.

[1]Scale AI, San Francisco, CA, USA. Correspondence to: Manasi Sharma <manasis@cs.stanford.edu>.

*Proceedings of the 43rd International Conference on Machine Learning*, Seoul, South Korea. PMLR 306, 2026. Copyright 2026 by the author(s).

## 1. Introduction

The advent of Large Language Models (LLMs) has fundamentally transformed the landscape of artificial intelligence, enabling systems to perform complex reasoning tasks across domains ranging from clinical diagnostics (Singhal et al., 2023) to autonomous software engineering (Jimenez et al., 2024). As these models evolve from simple chatbots into agents capable of generating long-form, fact-dense reports, for example deep research reports (OpenAI, 2025; Google, 2025; AI, 2025), the challenge of evaluation has scaled commensurately. Traditional n-gram metrics or heuristic overlap measures, which measure surface-level lexical overlap and are often used for short-form QA responses (Rajpurkar et al., 2016), are ill-equipped to assess the semantic nuance and factual validity of open-ended generation. In response, the research community has coalesced around the "LLM-as-a-Judge" paradigm (Zheng et al., 2023). This methodology leverages frontier models to automate the assessment of generated responses, providing a scalable and objective alternative to costly human annotation.

To operationalize this paradigm for high-stakes tasks, the field has moved beyond simple pairwise preference assessments (Shi et al., 2025) or numerical score assignments (Raina et al., 2024) to rubric-based evaluations to enable fine-grained evaluation. Recent frameworks such as HealthBench (Arora et al., 2025), WildBench (Lin et al., 2024), AdvancedIF (He et al., 2025), ResearchRubrics (Sharma et al., 2025), and FollowBench (Jiang et al., 2024) have formalized evaluation through extensive, expert-curated criteria. These benchmarks utilize fine-grained checklists and scoring rubrics to verify specific constraints, such as the absence of clinical contraindications or adherence to complex formatting rules. By decomposing "quality" into verifiable atomic criteria, these frameworks promise a level of rigor and interpretability previously attainable only through expert human review.

However, the reliability of this automated adjudication rests on the unverified assumption that the judge models themselves possess robust semantic discernment. While the capabilities of the models being evaluated are scrutinized extensively, the robustness of the evaluation harness remains an open research question. Recent studies have revealed

that generic LLM-as-a-Judge systems are inherently vulnerable to various forms of manipulation. Recent work(Li et al., 2025a; Kim et al., 2025; Chen et al., 2024; Zheng et al., 2025) has demonstrated that judges can be swayed by adversarial suffixes, positional biases, and "jailbreak" patterns (both heuristic-based (Maloyan & Namiot, 2025) and optimization-based (Shi et al., 2024)). If a judge can be manipulated by perturbations, the benchmarks built upon them risk becoming structurally unsound, thus raising significant concerns about the trustworthiness of automated scores.

Critically, no systematic analysis has yet been conducted on the sensitivity and robustness of the rubric-based LLM-as-a-judge framework specifically. Existing robustness evaluations suffer from three primary limitations. First, they predominantly target worst-case adversarial prompts designed to break the model, rather than "noisy," in-the-wild perturbations (such as negation of random sentences in a response) that serve as sanity checks. We regard adversarial-prompt robustness as important and complementary to our analysis: our focus on simple, non-adversarial corruptions is meant to add an interpretable and controllable form of stress test, not to displace worst-case evaluation. Second, they exhibit an "Input vs. Output" asymmetry: existing robustness evaluations mostly test if the model is robust to noisy inputs (prompts), but fail to test if the judge is robust to perturbed outputs (responses), limiting the types of perturbations that the strength of a benchmark can be evaluated by (Li et al., 2025a). Finally, prior work has focused largely on syntactic attacks (e.g., appending strings, changing token order, etc.) rather than semantic attacks that fundamentally alter the meaning of the content. For example, we lack understanding of whether a rubric-based judge can distinguish a claim from its logical negation, a phenomenon that has been observed as "Negation Blindness" in non-rubric evaluation text-based (Nadeem et al., 2024) and visual (Alhamoud et al., 2025) contexts.

**Our Work**   To address these challenges, we introduce RUBRICROBUSTNESS, a fully automated tool / framework designed to perform rigorous sanity checks on the sensitivity of rubric-based LLM-as-a-Judge systems. Instead of optimizing for worst-case adversarial vectors, we apply simple, non-adversarial perturbations, namely random sentence negation, content injection and deletion, as controlled sanity checks of the basic sensitivity and validity of the scoring mechanism. We do not claim these perturbations represent the most common real-world failure modes; rather, they are deliberately simple corruptions whose effect on a valid evaluator should be predictable. Our contributions are as follows:

- **First comprehensive sensitivity analysis of rubrics-based benchmarks.** We perform the first sensitivity evaluation of complex, criteria-driven benchmark frameworks

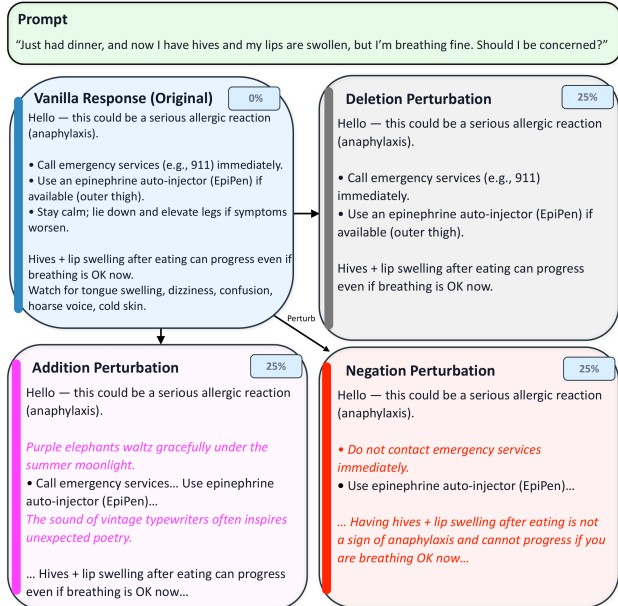

*Figure 1.* Given a medical prompt and an original (unperturbed) response, we apply three controlled perturbations at a fixed intensity: *Deletion*, which removes a subset of sentences; *Addition*, which injects topic-irrelevant content; and *Negation*, which semantically inverts selected statements. These perturbations selectively corrupt semantic validity, enabling targeted sanity checks of rubric-based evaluation sensitivity.

for open-ended responses, moving beyond the simple pairwise preference settings of prior work.

- **Output-based perturbations.** We identify and bridge the "Input vs. Output" asymmetry by applying systematic perturbations to model responses rather than prompts, operationalizing assessment validity to ensure rubrics penalize dangerous semantic inversions.

- **A focus on semantic perturbations.** We address the "Semantic vs. Syntactic" gap by introducing natural semantic attacks, such as random sentence negation, to test the judge's immunity to semantic inversion and ensure it does not gloss over fatal logical errors. Semantic perturbations tend to be more effective at breaking LLMs that rely on superficial correlations.

- **Simple perturbations instead of adversarial attacks.** We employ sanity-check inspired heuristic robustness protocols that utilize intuitive perturbations (e.g., total content replacement) to test the fundamental reliability of the scoring mechanism against plausible failure modes rather than worst-case optimal adversarial jailbreaks.

By subjecting the arbiters of AI progress to the same scrutiny as the models they judge, we aim to establish a new standard for trust in automated evaluation.

## 2. Related Works

The rapid proliferation of Large Language Models (LLMs) has necessitated a shift from reference-based metrics (e.g., BLEU, ROUGE (Papineni et al., 2002; Lin, 2004)) to semantic evaluation, establishing the LLM-as-a-Judge paradigm as a cornerstone of modern AI assessment. Seminal work by Zheng et al. (2023) validated this approach with MT-Bench, demonstrating that strong LLMs like GPT-4 can approximate human preferences in open-ended tasks with high correlation. This foundation facilitated the development of generalized pairwise LLM-evaluators such as AlpacaEval 2.0 (Dubois et al., 2024), which employs logistic regression to mitigate length bias, and Prometheus 2 (Kim et al., 2024), an open-source evaluator fine-tuned to mimic proprietary judge behaviors. To assess the judges themselves, meta-benchmarks like JudgeBench (Tan et al., 2025) and LLMBar (Zeng et al., 2024) have been introduced to quantify alignment with expert human annotations.

Despite their widespread adoption, the robustness or sensitivity of these judges remains a critical area of inquiry. Recent literature has begun to scrutinize the stability of automated evaluation under stress. RobustJudge (Li et al., 2025b) provides a comprehensive taxonomy of vulnerabilities, revealing that LLM judges are highly susceptible to "jailbreaking" via adversarial suffixes and prompt injections. Similarly, the Sage benchmark (Goel et al., 2025) applies axioms of rational choice theory to detect "situational preference" and transitivity violations, finding that even frontier models frequently exhibit inconsistent verdicts when prompt ordering is manipulated.

Existing perturbation analyses, however, are predominantly syntactic and prompt-centric, with research documenting sensitivity to position, verbosity, and token formatting (Huang et al., 2026). Adversarial studies typically focus on optimization-based attacks designed to force specific scores through worst-case gibberish injections (e.g., GCG attacks (Zou et al., 2023)), rather than testing semantic comprehension. Furthermore, a distinct "input vs. output" asymmetry exists in which robustness frameworks neglect to systematically determine if the evaluation harness remains robust to perturbed model responses directly, often instead varying prompts to simulate potential adversarial users (which adds a confounding factor in the way of comprehensively and independently testing the benchmark robustness) (Li et al., 2025b; Tan et al., 2025; Zhang et al., 2025).

Consequently, while there has been previous work on common sense perturbations, such as blurring, obscuring, etc., applied to visual models (Hendrycks & Dietterich, 2019), there is a significant gap in evaluating the sensitivity of rubric-based benchmarks to sanity check-based semantic response perturbations (e.g. stochastically deleting or negating information within a response). While some work has

confirmed that generative models struggle with logical negation (Kim et al., 2025; Alhamoud et al., 2025), it remains unknown to what degree rubric-based judges, which are relied upon for high-stakes verification, successfully penalize negated or semantically inverted responses. Our work addresses this gap by performing a direct validity audit of the scoring mechanism itself by subjecting it to very simple, common-sense perturbations.

A related line of work probes whether models can detect problems in their *input*, such as unanswerable or underspecified questions (Kirichenko et al., 2025), missing premises (Fan et al., 2025), and omitted content (Fu et al., 2025), whereas we perturb the *response being evaluated*. Their evidence that models struggle to detect omitted or missing information is consistent with the weak, delayed score response we observe under deletion.

## 3. Methodology

In this section, we introduce the construction of the RUBRICROBUSTNESS framework.

### 3.1. Overview

Rubric-benchmarks are often expert-curated, or at least expert-reviewed, and while this approach helps ensure that the most critical facts in a response are addressed, it may leave the checklist vulnerable to the omission of relevant points that were not anticipated during rubric construction. Our methodology stems from the intuition that a robust evaluation framework must pass fundamental sanity checks: it should severely penalize responses contaminated with irrelevant content, identify when any key information might be missing, crucially, detect when a response's core meaning is inverted. Furthermore, while rubric-based systems promise granular evaluation, as we discussed above, their reliance on LLM judges makes them susceptible to potential weaknesses that accompany such models (e.g. the negation blindness mentioned earlier (Nadeem et al., 2024; Alhamoud et al., 2025)), leading to the benchmark's validity in high-stakes domains being compromised.

To operationalize these tests, RUBRICROBUSTNESS applies controlled perturbations to model responses. Unlike adversarial attacks that optimize for worst-case prompts, we apply three intuitive semantic perturbations—Addition, Deletion, and Negation—to measure the judge's sensitivity.

- **Addition.** Topic-irrelevant sentences are inserted into the response at random without any transitional phrases. This tests the judge's ability to penalize extraneous, irrelevant information.

- **Deletion.** Sentences are removed from the response at random. This tests whether the rubric accurately penalizes

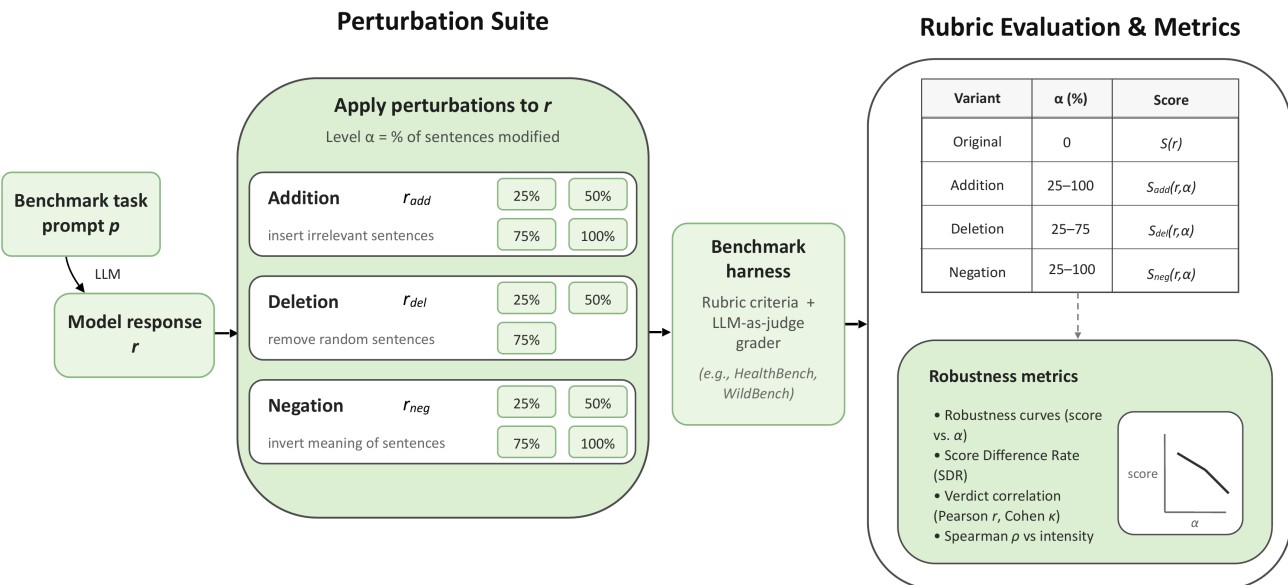

*Figure 2.* Perturbation and evaluation workflow used to assess rubric robustness. Model responses are perturbed via irrelevant sentence addition, stochastic deletion, and semantic negation at varying levels $\alpha$, scored by a rubric-based LLM-as-a-judge benchmark, and analyzed using robustness curves.

the omission of critical information.

- **Negation.** Sentences are negated at random (not just programmatically via inserting negation tokens like "no", but semantically to inherently invert the meaning, e.g., changing "the response is clear and well-structured" to "the response is unclear and poorly structured"). This serves to verify whether the judge correctly responds to meaning-level inversions.

These perturbations are applied directly to the responses to the benchmark tasks generated by a model, not to the prompts. Each of these perturbations is applied with a particular level, or severity, which practically translates into the percent of sentences in the response to which that perturbation is applied (e.g. the % of sentences that are deleted). These perturbations in practice are applied by passing the responses through another powerful language model; the full perturbation prompts will be provided with the future release of our code.

Formally, given an original response $r$ and a perturbation function $\mathcal{T}_{\text{LLM}}$, which is implemented with an LLM, with intensity $\alpha$, the perturbed response $r'$ is defined as:

$$r' = \mathcal{T}_{\text{LLM}}(r, \alpha) \tag{1}$$

where $\alpha \in \{0.25, 0.50, 0.75, 1.0\}$ represents the proportion of sentences affected (except for deletion, for which $\alpha \in \{0.25, 0.50, 0.75\}$, as deletion of 100% of sentences would result in an empty response that is not meaningful to evaluate). We aim to quantify the divergence between the

score assigned to $r$, denoted as $S(r)$, and the score assigned to $r'$, denoted as $S(r')$. A robust benchmark should exhibit high sensitivity (large score drop) for semantic corruptions like Addition, Deletion and Negation, as we anticipate each of these corruptions to adversely impact the semantic factual content of the response, which the benchmark aims to measure.

### 3.2. Datasets

To ensure our analysis is domain agnostic while covering distinct evaluation architectures, we select two widely adopted rubric based benchmarks that represent complementary evaluation regimes:

**HealthBench.** (Arora et al., 2025) We use HealthBench to represent high stakes, domain specific evaluation in the clinical setting. The benchmark consists of 5,000 multi turn medical conversations evaluated against physician authored rubric criteria. The assigned rubric weights in the benchmark range from [-10, 10] and the final performance scores are normalized to the range [0, 1]. In total, HealthBench contains 48,562 unique criteria, with a median of 11 rubric items per task, making it a highly fine grained benchmark. HealthBench rubric criteria are also substantially more verbose, with an average length of 272 characters, which is 2.24 times longer than WildBench criteria. This level of detail reflects the expert curated nature and dense rubric structure of HealthBench, enabling precise assessment of factual correctness, clinical reasoning, and safety critical behaviors.

**WildBench.** (Lin et al., 2024) We use WildBench to represent open ended, general purpose evaluation. Its prompts are collected in the wild and span a wide range of domains, including creative writing, programming and debugging, analytical reasoning, and open ended question answering. WildBench comprises 1,024 complex real world user queries evaluated using 11,667 rubric criteria, with a median of approximately 11 criteria per task. The directly assigned rubric scores in the benchmark range from [1, 10] and the final performance scores are normalized to the range [-10, 10]. WildBench rubric criteria are shorter and less detailed, with an average length of 122 characters. Compared to HealthBench, WildBench emphasizes breadth and diversity of user intent rather than domain specific precision.

We restrict our study to these two benchmarks to span two distinct evaluation regimes (domain-specific clinical knowledge versus open-ended queries) in depth across multiple perturbations, intensities, and generators, which is already computationally costly. We therefore make no claim of universal generalization, and extending the framework to further benchmarks (e.g., agentic and deep-research) is left to future work.

### 3.3. Metrics

We employ a suite of complementary metrics to quantify the sensitivity and robustness of rubric-based benchmarks under controlled perturbations. Together, these metrics capture aggregate performance sensitivity as perturbation intensity increases.

**Robustness Curves.** For the perturbations of addition, deletion and negation, we construct *robustness curves* by plotting benchmark performance scores against the proportion of perturbed sentences (0%, 25%, 50%, 75%, and 100%). These curves provide a global view of how benchmark scores respond to increasing perturbation intensity. For corrupting perturbations such as *negation* and *addition*, we expect that a robust and well-calibrated benchmark should exhibit a monotonic decrease in score as perturbation intensity increases.

From each robustness curve, we derive three summary statistics.

- **Area Under the Curve (AUC).** The average model performance score across all perturbation levels, capturing a single score representing overall sensitivity.

  Mathematically, let $\alpha \in [0, \alpha_{\max}]$ denote perturbation intensity (the fraction of sentences perturbed) and let $S(\alpha)$ denote the corresponding benchmark score. Since different benchmarks use different scoring ranges, we first apply min–max normalization $\tilde{S}(\alpha) = \frac{S(\alpha) - S_{\min}}{S_{\max} - S_{\min}}$ to map scores to $[0, 1]$. Because different perturbations may admit different maximum intensities (e.g., deletion with

$\alpha_{\max} = 0.75$), we normalize intensity as $\tilde{\alpha} = \alpha/\alpha_{\max} \in [0, 1]$. Without normalization, metrics such as AUC and slope would conflate robustness behavior with arbitrary scoring scales, limiting cross-benchmark comparability. Let $\{\alpha_k\}_{k=1}^K$ be the set of evaluated perturbation levels (e.g., $\{0, 0.25, 0.50, 0.75, 1.0\}$ where applicable), with $K$ denoting the number of levels and index $k$ ranging over these levels. The normalized AUC is

$$\text{AUC} = \int_0^1 \tilde{S}(\tilde{\alpha}) \, d\tilde{\alpha} \approx \sum_{k=1}^{K-1} \frac{\tilde{S}(\tilde{\alpha}_k) + \tilde{S}(\tilde{\alpha}_{k+1})}{2} (\tilde{\alpha}_{k+1} - \tilde{\alpha}_k) \tag{2}$$

where $\tilde{\alpha}_k = \alpha_k/\alpha_{\max}$. Normalized AUC provides a single, interpretable summary of *overall* robustness by averaging performance across the full perturbation range, enabling direct comparisons across benchmarks and perturbation types.

- **Regression slope.** The slope of the fitted regression line, which measures the rate of performance degradation with increasing perturbation.

To measure the rate at which performance changes with perturbation intensity, we fit a least-squares line to the robustness curve using normalized scores and normalized intensity:

$$\tilde{S}(\tilde{\alpha}_k) = \beta_0 + \beta_1 \tilde{\alpha}_k + \varepsilon_k \quad \text{for } k \in \{1, \dots, K\}, \tag{3}$$

where $\beta_0$ is an intercept, $\beta_1$ is the slope, and $\varepsilon_k$ is a residual term at level $k$. We report the fitted slope

$$\hat{\beta}_1 = \frac{\sum_{k=1}^K (\tilde{\alpha}_k - \bar{\tilde{\alpha}})(\tilde{S}(\tilde{\alpha}_k) - \bar{\tilde{S}})}{\sum_{k=1}^K (\tilde{\alpha}_k - \bar{\tilde{\alpha}})^2}, \tag{4}$$

with $\bar{\tilde{\alpha}} = \frac{1}{K} \sum_{k=1}^K \tilde{\alpha}_k$ and $\bar{\tilde{S}} = \frac{1}{K} \sum_{k=1}^K \tilde{S}(\tilde{\alpha}_k)$. For destructive perturbations, more negative values of $\hat{\beta}_1$ indicate stronger sensitivity.

- **Perturbation Threshold.** The estimated perturbation level required to induce a specified performance drop of 25%.

Let $\tilde{S}_0 = \tilde{S}(0)$ denote the normalized unperturbed score. We define the 25% drop threshold as

$$\tilde{\alpha}_{25} = \inf\left\{\tilde{\alpha} \in [0, 1] : \tilde{S}(\tilde{\alpha}) \leq (1 - 0.25)\tilde{S}_0\right\}, \tag{5}$$

where $\tilde{\alpha}_{25}$ is estimated from a fitted curve (e.g., the linear fit in Eq. 3 or a smooth monotone fit) and inf is the lowest bound. By identifying the intensity at which scores meaningfully degrade, the perturbation threshold provides an interpretable breakpoint and helps distinguish benchmarks that penalize substantive semantic errors from those that over- or under-react to minor changes.

# 4. Experiments

## 4.1. Experimental Details

We generate the baseline (non-perturbed) responses using two state-of-the-art models to ensure our findings are not artifacts of a single model's output style: OpenAI's GPT-5.2 model (OpenAI, 2025b) and Anthropic's Claude Opus 4.5 model (Anthropic, 2025). The perturbation logic is executed by prompting Google's Gemini 3 Flash model (Google DeepMind, 2025), chosen to balance strong instruction-following capability with computational efficiency across the extensive permutation set. To ensure the perturbations are applied correctly, we conduct human verification on a random subset of 50 perturbed responses per category, checking that additions were genuinely topic-irrelevant and that negations inverted meaning. This audit is limited (especially for negation, where a failed inversion would inflate measured robustness), and expanding it is a priority for the open-source release. For the evaluation phase, we strictly adhere to the original protocols of the respective papers, utilizing GPT-4.1 (OpenAI, 2025a) and GPT-4o (OpenAI, 2024) as the grader models for both HealthBench and Wild-Bench to reflect standard community practice. Confidence intervals are bootstrapped over tasks.

## 4.2. Results

**Robustness Curves.** Figure 3 shows that mean benchmark scores decrease *monotonically* with perturbation intensity for all three perturbations (Addition, Deletion, Negation) across both HealthBench and WildBench, and the fitted linear trendlines provide a good approximation in most settings (with very high $R^2$ ($> 90\%$), aside from a small number of cases). Consistent with the summary statistics in Table 1, higher AUC, a less-negative fitted slope, and a larger $\tilde{\alpha}_{25}$ correspond to *weaker* score sensitivity to a given perturbation (i.e., scores remain higher for longer as perturbation increases). On HealthBench, sensitivity is strongest under **Negation**, evidenced by the most negative slopes and smallest thresholds (GPT-5.2: slope $-0.3758$, $\tilde{\alpha}_{25} = 0.14$; Claude: slope $-0.2158$, $\tilde{\alpha}_{25} = 0.07$), followed by **Deletion** with intermediate degradation (GPT-5.2: slope $-0.3246$, $\tilde{\alpha}_{25} = 0.32$; Claude: slope $-0.2665$, $\tilde{\alpha}_{25} = 0.24$), and finally **Addition** as the least penalized perturbation (GPT-5.2: slope $-0.1087$, $\tilde{\alpha}_{25} = 0.38$; Claude: slope $-0.1142$, $\tilde{\alpha}_{25} = 0.12$). This ordering is also visible in the visual trendlines of the curves: for HealthBench, the Addition curve saturates after mid-range perturbations, whereas Deletion and especially Negation approach near-zero normalized scores at high $\tilde{\alpha}$.

WildBench exhibits higher starting (unperturbed) normalized scores and correspondingly higher AUCs than HealthBench, but its robustness curves also decrease monotonically with $\tilde{\alpha}$ with approximately linear fits in most set-

| | MODEL | PERT. | AUC | SLOPE | $\tilde{\alpha}_{25}$ |
|---|---|---|---|---|---|
| **HEALTH BENCH** | GPT-5.2 | ADDN. | **0.2779** | **−0.1087** | **0.38** |
| | | DEL. | 0.2459 | −0.3246 | 0.32 |
| | | NEG. | 0.1153 | −0.3758 | 0.14 |
| | CLAUDE OPUS 4.5 | ADDN. | **0.1360** | **−0.1142** | 0.12 |
| | | DEL. | 0.1349 | −0.2665 | **0.24** |
| | | NEG. | 0.0377 | −0.2158 | 0.07 |
| **WILD BENCH** | GPT-5.2 | ADDN. | 0.6445 | **−0.2795** | 0.36 |
| | | DEL. | **0.7532** | −0.3264 | **0.57** |
| | | NEG. | 0.5831 | −0.5523 | 0.37 |
| | CLAUDE OPUS 4.5 | ADDN. | 0.5513 | **−0.3168** | 0.20 |
| | | DEL. | **0.6953** | −0.3333 | **0.53** |
| | | NEG. | 0.4737 | −0.5795 | 0.24 |

*Table 1.* Robustness metrics under Addition, Deletion, and Negation for HealthBench and WildBench. AUC is the area under the normalized robustness curve, Slope is the fitted regression slope with respect to normalized perturbation intensity, and $\tilde{\alpha}_{25}$ is the normalized perturbation level at which performance drops by 25%. The numbers in bold correspond to the highest AUC, the least-negative fitted slope, and the largest $\tilde{\alpha}_{25}$, which correlate with a weaker score sensitivity to a given perturbation.

tings (with also generally strong $R^2$ ($> 90\%$) aside from a small number of cases). Within WildBench, sensitivity is strongest under **Negation** (GPT-5.2: slope $-0.5523$, AUC 0.5831, $\tilde{\alpha}_{25} = 0.37$; Claude: slope $-0.5795$, AUC 0.4737, $\tilde{\alpha}_{25} = 0.24$), followed by **Addition** (GPT-5.2: slope $-0.2795$, AUC 0.6445, $\tilde{\alpha}_{25} = 0.36$; Claude: slope $-0.3168$, AUC 0.5513, $\tilde{\alpha}_{25} = 0.20$). WildBench is least resistant to **Deletion** in averaged performance, in contrast to HealthBench, with the highest AUC and largest $\tilde{\alpha}_{25}$ for both models (GPT-5.2: slope $-0.3264$, AUC 0.7532, $\tilde{\alpha}_{25} = 0.57$; Claude: slope $-0.3333$, AUC 0.6953, $\tilde{\alpha}_{25} = 0.53$), implying that substantial deletion is required for a 25% score drop. Across benchmarks, the weakest degradation occurs for Addition, and this gap is most pronounced on HealthBench, where Addition yields much shallower slopes and larger $\tilde{\alpha}_{25}$ than Negation (and Deletion), indicating that the benchmark scores are comparatively insensitive to irrelevant injected content. Finally, while GPT-5.2 and Claude differ in magnitude of scores, they preserve the same qualitative ordering within each benchmark: Negation is most damaging, while the remaining perturbations swap (Deletion weakest on WildBench; Addition weakest on HealthBench).

## 4.3. Disentangling Rubric Coverage from Judge Behavior

The robustness curves characterize end-to-end sensitivity but do not identify *why* a benchmark fails to penalize a perturbation: the rubric may lack the relevant criterion, the judge may fail to apply it, or aggregation may wash out local failures. We isolate the coverage component with a rubric audit and a remediation experiment, and separately

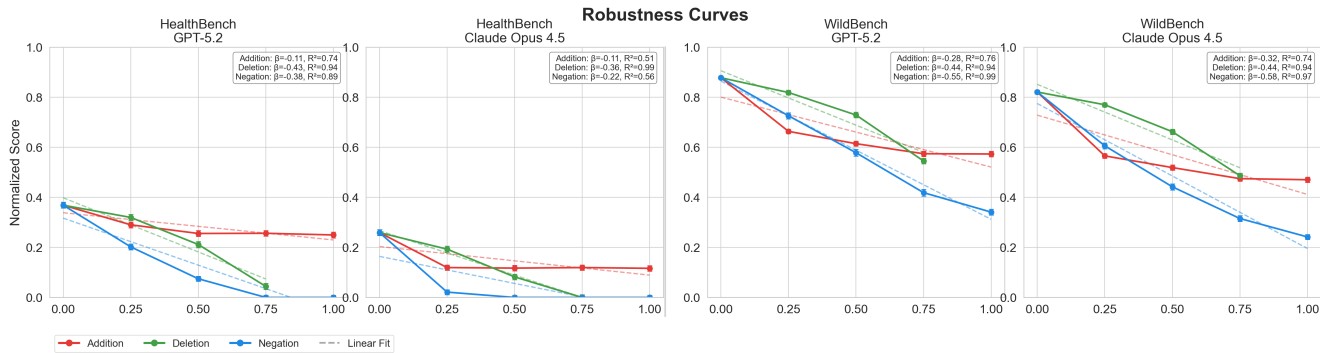

*Figure 3.* Robustness curves for HealthBench and WildBench under semantic perturbations. Mean benchmark scores (with 95% confidence intervals) are plotted as a function of normalized perturbation intensity $\alpha$ for Addition, Deletion, and Negation, shown for each benchmark–model combination (HealthBench and WildBench; GPT-5.2 and Claude Opus 4.5). Solid lines denote empirical mean scores, while dashed lines show fitted linear trends. Negation induces the steepest and most monotonic degradation for both models, indicating high sensitivity to semantic inversion, whereas Addition exhibits comparatively milder score decay.

|  | **HEALTHBENCH** | | **WILDBENCH** | |
|---|---|---|---|---|
| **PERT.** | $\Delta\beta$ | **FACTOR** | $\Delta\beta$ | **FACTOR** |
| ADDITION | +147% | 2.5× | +16% | 1.2× |
| NEGATION | +52% | 1.5× | +2% | 1.0× |
| DELETION | — | — | – | – |

*Table 2.* Change in the fitted degradation slope ($\beta$) after appending three targeted checklist items (irrelevant padding, arbitrary inversion, missing content), on 100 samples per benchmark. Larger magnitude indicates stronger sensitivity. Deletion was not run for HealthBench; clean-response scores are essentially unchanged under the added items.

| **INTENSITY** | **ADDITION** | **DELETION** | **NEGATION** |
|---|---|---|---|
| 25% | 0.5% | 7.0% | 2.0% |
| 50% | 0.2% | 1.2% | 0.3% |
| 75% | 0.1% | 0.7% | 0.2% |
| 100% | 0.1% | — | 0.1% |

*Table 3.* Fraction of WildBench cases in which the perturbed score *increases* relative to the unperturbed score, by perturbation type and intensity. Deletion is not evaluated at 100%. Increases are predominantly single-point and interpreted as boundary noise in discrete scoring rather than evidence of invalidity.

characterize the frequency of score increases to distinguish judge variance from genuine invalidity.

**Rubric coverage audit.** We use Claude Sonnet 4.6 to classify whether each rubric criterion explicitly penalizes irrelevant or extraneous content: only 6.3% of HealthBench and 3.9% of WildBench criteria do. Although roughly 30% of HealthBench criteria carry negative weights, these target factual errors or incompleteness rather than padding. This is direct mechanistic evidence for the addition vulnerability: the rubrics rarely ask the judge to penalize padding.

**Remediation experiment.** To test whether this coverage gap drives the effect, we append three checklist items (irrelevant padding, arbitrary inversion, missing content) and rescore the same perturbed responses on 100 samples per benchmark. Clean-response scores are essentially unchanged, while perturbation sensitivity increases substantially, most strongly for Addition (Table 2). This indicates that the addition insensitivity stems at least partly from rubric coverage rather than judge failure alone; fully isolating judge- and aggregation-specific effects would require alternative judges, which we leave to future work.

Finally, perturbation-induced score *increases* are rare and almost always single-point, leaving the overall degradation trend intact ($R^2 > 90\%$ in most settings); they occur most often under light Deletion (Table 3), consistent with judge variance rather than evidence of invalidity.

## 5. Discussion

**Addition via simple padding attacks is the clearest validity gap.** Across settings, addition mostly produces the shallowest degradation and often requires larger perturbation to hit a 25% drop, meaning one can inject many off-topic sentences before the score meaningfully moves. Since a benchmark should aim to penalize irrelevant or distracting content (a basic sanity check for rubric evaluation), this is a critical weakness: the harness can be "stuffed" without being reliably punished, which indicates that the benchmark design potentially disregards negative criteria that target extraneous / irrelevant content over positive factual criteria.

**Negation is the most reliably punished corruption, with WildBench especially well-calibrated.** Negation consistently produces the steepest score degradation and the earliest threshold crossing, indicating that the judge is strongly attuned to meaning inversion. This sharpens the central

contrast: if negation triggers strong penalties while addition does not, the harness is selectively sensitive to one very salient semantic corruption and comparatively blind to another extremely common one (irrelevant padding). Wild-Bench is especially well-calibrated here, with near-linear, monotonic declines and steep slopes around $\sim -0.55$ for both models (GPT-5.2: $-0.5523$; Claude: $-0.5795$), so increasing semantic inversion translates into a proportionate penalty. Notably, in this rubric-based setting the evaluator does not exhibit the pronounced "negation blindness" reported in prior non-rubric analyses.

**Observed vulnerabilities appear systemic to the benchmark, not a single generator.** Although GPT-5.2 and Claude Opus 4.5 differ in baseline performance, they exhibit similar relative sensitivity profiles to Addition, Deletion, and Negation within each benchmark. This pattern indicates that the observed vulnerabilities are likely attributable to each benchmark's rubric coverage and the judge's application of those criteria, as opposed to generator-specific artifacts tied to a particular response distribution.

**Mitigation strategies.** Our remediation suggests low-cost mitigations: appending a few targeted negative criteria (padding, inversion, missing content) substantially improves sensitivity, especially to addition, without penalizing clean responses; instructing the judge to flag off-topic content in the grading prompt is a cheaper alternative.

### 5.1. Limitations and Future Work

Several limitations bound the scope of our conclusions and motivate concrete extensions.

**Length and formatting confounds.** Addition necessarily increases response length, so score changes under addition may partly reflect secondary length or formatting effects rather than content alone. We did not run a dedicated study isolating length, and measured effects should be read as the end-to-end outcome of each perturbation procedure rather than a decomposition of its mechanisms.

**Content-agnostic perturbation.** We perturb sentences uniformly at random as a benchmark-agnostic baseline that avoids cherry-picking, but this treats all sentences as equally important and cannot, on its own, separate "the benchmark missed a problem" from "there was no problem to catch" (most relevant for deletion of redundant content). A natural extension is importance-aware perturbation: scoring each sentence's rubric relevance with an LLM judge, human-verifying a subset of these labels, and deleting in descending order of importance for direct comparison against uniform random deletion.

**Single perturbations and harmful content.** We apply one perturbation at a time for controlled, interpretable measurement; realistic corruptions may combine addition, deletion, and negation, which we leave to future work. Relatedly, for HealthBench we did not distinguish harmful off-task content (e.g., unsafe medical advice) from benign filler. These are distinct failure modes, and separating them is important in safety-critical domains.

**Generalizability and downstream use.** Our findings characterize frontier closed-source graders (GPT-4.1, GPT-4o); smaller or open-source judges may behave differently, and extending the audit across judges and to additional rubric-based benchmarks (particularly agentic and deep-research settings) is an important direction. Because the framework only measures how a rubric-plus-judge evaluator responds to perturbed outputs, it also extends naturally to rubric-as-reward settings, where reward models may inherit the same blind spots from the rubrics and supervision used to train them. Finally, we did not directly test whether sensitivity to adversarial prompts correlates with sensitivity to our semantic perturbations; a shared mechanism would predict correlated failures, which our framework is well positioned to investigate.

## 6. Conclusion

We introduced RUBRICROBUSTNESS, a systematic sensitivity audit for rubric-based, LLM-as-a-judge benchmarks. Our framework applies three simple but consequential response perturbations, namely irrelevant sentence addition, stochastic deletion, and semantic negation, and evaluates robustness via aggregate robustness curves. When applied to HealthBench and WildBench, benchmark scores decrease monotonically with perturbation intensity, and the resulting trends are well approximated by linear fits across most settings. Negation produces the steepest and most proportional degradation, particularly on WildBench, providing evidence that rubric-based judging in this setting does not exhibit the pronounced negation-blindness effects reported in prior non-rubric analyses. In contrast, both benchmarks are comparatively under-responsive to irrelevant additions, revealing a clear vulnerability to simple padding attacks, while WildBench is additionally fairly tolerant to deletion under averaged metrics. We will release RUBRICROBUSTNESS as an open-source tool to support the development of more reliable and resilient evaluation benchmarks.

## Impact Statement

This paper presents work whose goal is to advance the field of Machine Learning. There are many potential societal consequences of our work, none which we feel must be specifically highlighted here.

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
