# OpenReview forum: "RubricRobustness: Evaluating the Sensitivity of Rubrics-Based Benchmarks to Simple Perturbations"
_ICML.cc/2026/Conference — ICML 2026 regular_

### Official Review · Reviewer_EXuc · 2026-03-09

**Soundness:** 3
**Presentation:** 2
**Significance:** 2
**Originality:** 2
**Overall Recommendation:** 2
**Confidence:** 5

**Summary:**

This work performs a meta-evaluation on rubric-based evaluation protocols. The main contribution is an automated framework that runs semantic perturbations (negate certain statements, delete sentences, or add irrelevant content), and checks whether the scoring protocol can catch the injected ill behavior. The authors run analyses on two benchmarks: HealthBench and WildBench, and find that negation causes score drop sharply, while addition has a much softer effect.

**Compliance With Llm Reviewing Policy:**

Affirmed.

**Final Justification:**

The authors propose an interesting perturbation-based framework for analyzing rubric-based evaluation protocols, with control over perturbation severity and empirical analysis on two benchmarks. However, the study is limited to only two benchmarks despite positioning itself as a general framework, and the claim that semantic perturbation sensitivity is a necessary prerequisite for validating rubric coverage is not well supported. I also found parts of the writing difficult to follow.

In the rebuttal, the authors narrowed some of the claims and provided more intuition about semantic perturbations as a diagnostic tool. However, my main concern is still unresolved. The paper continues to assume that sensitivity to addition, deletion, and negation is strong evidence of rubric coverage or benchmark validity. It also remains unclear how perturbation sensitivity connects to downstream evaluation reliability or alignment with task-relevant quality. The paper is still limited by only evaluating two benchmarks, and I was not fully convinced by the discussion of why adversarial prompt robustness is treated as less central than these perturbation tests. I will maintain my current score.

Overall, my recommdntation is Reject.

**Key Questions For Authors:**

1. Why is targeting “worst-case adversarial prompts” considered a limitation (line 071)? It’s a realistic risk with how people would want to game LLM judge systems. In practice, it feels less likely that people would do random addition, deletion, or negation in their model output.
2. Is there a correlation between sensitivity to adversarial prompts and sensitivity to semantic perturbations?
3. What would be some reasonable strategies to improve robustness? What happens if you add a lot of general quality-oriented checks in the rubrics?
4. Not a question but a small comment: when typing double quotes, the starting quote should be `` in latex, otherwise it renders as a closing quote.

**Limitations:**

yes

**Strengths And Weaknesses:**

# Strengths
1. The focus on perturbing outputs is interesting, and provides an alternative perspective to common literature that looks at prompt sensitivity.
2. The authors enforce careful control of perturbation severity.
3. The statistical experiments are rigorous, and the authors extensively analyze the results.

# Weaknesses
1. The experiments and analyses are limited to only two benchmarks. If the goal is to propose a systematic framework for rubric-based evaluation protocols more broadly, the authors should consider scaling the study to additional benchmarks (e.g., benchmarks for deep research agents).
2. The core question is whether the rubric can truly capture qualities that matter for the given task. While the authors claim that semantic perturbation is a necessary prerequisite for validating rubric coverage (line 042), the paper does not convincingly support this claim.
3. The writing sometimes feels redundant, under-motivated, or hard to follow. For example, the “Verdict Consistency” paragraph has a lot of vague wording that is really hard to understand. It’s hard to decipher what “distinguish coherent score shifts from instability driven by criterion-level decision flips that may be masked by weighted aggregation” means exactly.

---

> ### Author Rebuttal · Authors · 2026-03-31
>
> We thank the reviewer for the detailed and constructive feedback. We respond to each of the concerns below and will incorporate these clarifications in the final manuscript:
>
> 1. “The experiments and analyses are limited to only two benchmarks”
> We agree broader benchmark coverage would strengthen the paper. HealthBench and WildBench were chosen to represent very distinct evaluation regimes: domain-specific clinical knowledge versus open-ended general queries, with different rubric architectures and scoring ranges, and evaluating both in depth across multiple perturbation types and models was already very costly. We will make this scope decision explicit, ensure no overgeneralization claims are being made, and flag additional benchmarks, especially agentic or deep-research benchmarks, as future work.
>
> 2. “While the authors claim that semantic perturbation is a necessary prerequisite for validating rubric coverage (line 042), the paper does not convincingly support this claim.”
> We agree that this claim should be stated more carefully. Our intent was not to argue that perturbation sensitivity alone is sufficient to validate rubric coverage, nor that it fully determines whether a rubric captures all task-relevant qualities. Rather, we think that sensitivity to simple semantic failures is a useful and, we would argue, important diagnostic check for rubric-based evaluators (e.g. like how unit tests are a necessary prerequisite for software reliability without guaranteeing correctness).
>
> The empirical case rests on our finding that both benchmarks fail even these simple checks: addition insensitivity reveals that a response can be substantially degraded without much score consequence, which is a basic validity failure by any reasonable definition of a quality benchmark. Our rubric criteria audit demonstrates that fewer than 7% of criteria explicitly penalize extraneous content on the both benchmarks and this provides direct mechanistic evidence: the benchmarks structurally cannot penalize padding because the rubrics never ask them to.
> We will sharpen this framing in the revision to make the argument more precise. We will revise the paper to narrow this wording and present perturbation analysis as one practical component of benchmark validation, rather than a complete test of rubric adequacy.
>
> 3. “Why is targeting “worst-case adversarial prompts” considered a limitation (line 071)?”
> We do not mean that adversarial-prompt robustness is unimportant. We agree it is highly relevant, especially when evaluators may be gamed in deployment. Our point was narrower: focusing only on worst-case adversarial attacks can make it harder to understand how a benchmark behaves under simpler, non-adversarial semantic corruptions that are more interpretable and controllable kinds of stress tests. We see these two settings as complementary, not competing. In revision, we will clarify this framing so it does not read as dismissive of adversarial evaluation.
>
> 4. “Is there a correlation between sensitivity to adversarial prompts and sensitivity to semantic perturbations?”
> We did not test this directly, and we agree it is a genuinely interesting empirical question. We would also be able to look into this as a natural future experiment, as if benchmarks sensitive to adversarial attacks are also sensitive to our semantic perturbations, it would suggest a shared underlying mechanism, but if not, it would imply the two distinct failure modes. Our framework is well positioned to extend this in future work.
>
> 5. “What would be some reasonable strategies to improve robustness? What happens if you add a lot of general quality-oriented checks in the rubrics?”
> We thank the reviewer for posing this question, and we think that at the granular rubric and judge level:
> We run a small remediation experiment on 100 samples from HealthBench and 100 from WildBench. We append three checklist items targeting padding, arbitrary inversion, and missing content, then rescore the same perturbed responses through the original judge pipeline. Clean-response scores remain essentially unchanged, indicating that the added items do not penalize unperturbed outputs. In contrast, perturbation sensitivity improves substantially, especially for addition:
> These results suggest that at least part of the observed addition insensitivity is due to rubric coverage rather than judge failure alone, and adding rubrics to target each of these cases can help, especially in the case of addition.
> Additionally, at the judge prompt level, we could try explicitly instructing the judge to flag and penalize off-topic additions, as opposed to having it repeated at the rubric-level for a lower-cost intervention.
> At a high level, users could also just use RubricRobustness out-of-the-box as a standard validity check before deploying any new benchmark, analogous to unit testing.
>
> Q4: LaTeX quote formatting
> We thank the reviewer for flagging this, and we will fix all instances in the revision.

---

> > ### Author Rebuttal · Reviewer_EXuc · 2026-04-03
> >
> > Thank you for the clarifications in the rebuttal. I appreciate the authors narrowing some of the claims and providing more intuition about semantic perturbations as a diagnostic tool. However, my main concern is still unresolved. The paper continues to assume that sensitivity to addition, deletion, and negation is strong evidence of rubric coverage or benchmark validity. It also remains unclear how perturbation sensitivity connects to downstream evaluation reliability or alignment with task-relevant quality. The paper is still limited by only evaluating two benchmarks, and I was not fully convinced by the discussion of why adversarial prompt robustness is treated as less central than these perturbation tests. I will maintain my current score.

---

> > > ### Author Response · Authors · 2026-04-08
> > >
> > > Thank you for the response.

---

### Official Review · Reviewer_iXxu · 2026-03-13

**Soundness:** 3
**Presentation:** 3
**Significance:** 3
**Originality:** 3
**Overall Recommendation:** 4
**Confidence:** 4

**Summary:**

This work introduces RubricRobustness, a systematic sensitivity audit for rubric-based, LLM-as-a-judge benchmarks. RubricRobustness applies three simple but consequential response perturbations, namely irrelevant sentence addition, stochastic deletion, and semantic negation, and evaluates robustness via both aggregate robustness curves and rubriclevel verdict consistency. This work investigates the extent to which manipulating the semantic veracity of a model’s response impacts its resulting score by applying the robustness framework to two of the most popular rubrics-based benchmarks: HealthBench and WildBench. Findings reveal systematic vulnerabilities. The authors argue that perturbation-based sensitivity analyses of this form are a necessary prerequisite for validating rubric coverage, ensuring that automated evaluation frameworks reliably penalize basic semantic failures.

**Compliance With Llm Reviewing Policy:**

Affirmed.

**Final Justification:**

Part of my concerns has resolved but I still have concern about generalizability across the broader LLM-as-a-judge ecosystem. So I decide to keep my score.

**Key Questions For Authors:**

1. Based on results observed by this paper, what would be a good mitigation strategy for users?
2. Would this method adaptable to Rubric-as-Reward models?

**Limitations:**

It may be helpful to discuss the generalizability across the broader LLM-as-a-judge ecosystem, as the sensitivity to rubrics and perturbations can vary significantly between the frontier closed-source models tested and smaller or open-source alternatives.

**Strengths And Weaknesses:**

Strengths
- The research question is well-motivated, interesting and important
- The perturbation methodology is conceptually clear and well-defined
- Overall, the paper is well written and easy to follow. And it incorporates figures that are helpful for understanding this work.

Weaknesses
- The framework identifies vulnerabilities but does not propose concrete strategies to: redesign rubrics, modify judge prompts or mitigate padding vulnerabilities.
- Testing more models would be helpful for the solidity of the findings

typos:
- ”Semantic vs. Syntactic”  --> ""
- Contribution 4: Simple Perturbations instead of adversarial attacks. --> Simple perturbations
- Citation format does not align with common academic practivce. eg: in related work, Seminal works by (Zheng et al., 2023) validated this
- Section 5, first paragraph: Across settings, Addition mostly -->  addition

---

> ### Author Rebuttal · Authors · 2026-03-31
>
> We are grateful for the careful and constructive review. We respond to each comment below, and all clarifications will be reflected in the final manuscript:
>
> 1. “Based on results observed by this paper, what would be a good mitigation strategy for users?”
> We thank the reviewer for posing this question, and we think that at the granular rubric and judge level:
> First, we conduct a rubric audit to isolate the coverage component. We used Claude Sonnet 4.6 to classify each rubric criterion/checklist item in HealthBench and WildBench according to whether it explicitly penalizes irrelevant or extraneous content. By this analysis, only 6.3% of HealthBench criteria and 3.9% of WildBench criteria do so. While roughly 30% of HealthBench criteria have negative weights, they mostly target factual correctness or incompleteness in otherwise relevant content rather than extraneous padding.
> Second, we run a small remediation experiment on 100 samples from HealthBench and 100 from WildBench. We append three checklist items targeting padding, arbitrary inversion, and missing content, then rescore the same perturbed responses through the original judge pipeline. Clean-response scores remain essentially unchanged, indicating that the added items do not penalize unperturbed outputs. In contrast, perturbation sensitivity improves substantially, especially for addition:
> These results suggest that at least part of the observed addition insensitivity is due to rubric coverage rather than judge failure alone, and adding rubrics to target each of these cases can help, especially in the case of addition.
> Additionally, at the judge prompt level, we could try explicitly instructing the judge to flag and penalize off-topic additions, as opposed to having it repeated at the rubric-level for a lower-cost intervention.
> At a high level, users could also just use RubricRobustness out-of-the-box as a standard validity check before deploying any new benchmark, analogous to unit testing.
>
> 2. “Would this method adaptable to Rubric-as-Reward models?”
> Yes, we believe that it would be well-suited. Our framework is agnostic to the particular scoring mechanism as it only measures how a rubric + judge evaluator responds to controlled perturbations of candidate outputs. For that reason, the same procedure should extend naturally to rubric-as-reward settings, where one would examine how the reward changes under addition, deletion, or negation perturbations. We think this is a promising direction, since reward models may inherit similar sensitivities or blind spots from the rubrics and supervision used to train them. We will add a brief discussion of this broader applicability.
>
> 3. “Generalizability across the broader LLM-as-a-judge ecosystem”
> We acknowledge this is a real limitation. Our findings characterize frontier closed-source judges (GPT-4.1, GPT-4o), and sensitivity patterns may differ substantially for smaller or open-source alternatives (e.g. smaller models may potentially exhibit even stronger negation blindness). We will add an explicit discussion of this in the limitations section and note it as a priority for the open-source release.
>
> Typos
> We thank the reviewer for catching all these errors and will fix all four in the revision.

---

> > ### Author Rebuttal · Reviewer_iXxu · 2026-04-03
> >
> > Thanks the authors for adding experiments on the mitigation question and for justifying the broader applicability. My only remaining concern is the generalizability across the broader LLM-as-a-judge ecosystem.

---

> > > ### Author Response · Authors · 2026-04-08
> > >
> > > Thank you for the response.

---

### Official Review · Reviewer_wmAy · 2026-03-13

**Soundness:** 2
**Presentation:** 3
**Significance:** 3
**Originality:** 3
**Overall Recommendation:** 4
**Confidence:** 2

**Summary:**

This paper proposes a systematic sensitivity analysis framework called RubricRobustness to audit a criterion-based LLM-as-judge benchmark under simple, common-sense output perturbations. The core idea is to perturb the model response in three ways: (i) semantic negation, (ii) random deletion, and (iii) irrelevant addition, varying the perturbation strength. Changes in benchmark scores and criterion-level judgments are then measured. Robustness is measured using robustness curves, AUC, the slope of the fit degradation, a 25% descent threshold, and judgment consistency metrics. They applied this framework to two popular rating-based benchmarks—HealthBench and WildBench—using responses from two powerful generators (GPT-5.2 and Claude Opus 4.5), perturbations generated by Gemini 3 Flash, and scores from the benchmarks' standard LLM raters (GPT-4.1 / GPT-4o). The key finding was that these benchmarks are highly sensitive to semantic negation but relatively less sensitive to irrelevant additions, suggesting a validity deficiency in the rating criteria/raters' ability to address fundamental semantic failure patterns.

**Compliance With Llm Reviewing Policy:**

Affirmed.

**Final Justification:**

The idea is interesting, I've decided to keep my score, weak accept, but I only have such a low confidence of 2.

**Key Questions For Authors:**

1. What are the specific prompts used to generate the add/delete/negate actions? How can you (quantitatively) ensure that (i) the add action is truly irrelevant to the topic and does not add content relevant to the scoring criteria; (ii) the delete action is uniformly randomized within the sentence; and (iii) the negation action reverses the meaning of the sentence and does not introduce other factors (style/format/length variations, contradictions, meaningless content)?
2. How frequently do scores increase under delete/negate/add actions? Do you consider these cases noise, expected behavior, or evidence of inconsistency between scoring criteria/judges? Detailed analysis will influence my understanding of "sensitivity" versus "invalidity."
3. Can you distinguish whether these deficiencies primarily stem from the rater model or the scoring criterion design? For example, do similar sensitivity patterns appear when the same perturbation response is scored by another rater?

**Limitations:**

No. The paper should address more explicitly the following limitations: (i) potential confusion caused by length/formatting issues introduced by perturbations; (ii) situations where perturbations can reasonably improve rating criterion satisfaction (non-monotonicity) etc.

**Strengths And Weaknesses:**

Strengths:
1. This paper focuses on an important but under-examined aspect that the evaluation framework itself including scoring criteria and judges, not just the model's robustness to prompt noise.
2. The perturbations are designed to be concise and easy to understand; the metrics are reasonable and complementary, distinguishing score drift from decision instability.
3. The experimental design reasonably avoids single-model artifacts by using two response generators and a canonical judge of the benchmark.
4. Including some human validation to verify the correctness of the perturbations is a good practice.

Weaknesses:
1. This framework implicitly assumes that addition/deletion/negation operations exhibit a monotonically decreasing trend; however, scoring criteria may not be monotonically decreasing, e.g., deleting harmful clauses may improve safety, adding warnings may improve clinical recommendations. Papers should handle such cases more carefully and quantify their frequency and whether they have been filtered.
2. Causal attribution remains somewhat ambiguous that observed sensitivity may reflect (a) flawed scoring criteria, (b) interactions with format/length effects, or (c) benchmark aggregation design. This paper describes this as “benchmark vulnerability,” but a clearer decomposition/ablation analysis would more strongly support this assertion.

---

> ### Author Rebuttal · Authors · 2026-03-31
>
> We thank the reviewer for the detailed feedback. We address each point below and will incorporate the relevant changes in the final version:
>
> 1. “What are the specific prompts used to generate the add/delete/negate actions”
> Full prompts will be included in the appendix. A brief summary is: addition prompts Gemini 3 Flash in detail to generate sentences from a completely unrelated domain inserted at random positions without transitional phrases, deletion selects sentences uniformly at random, negation instructs semantic meaning inversion while preserving sentence structure [examples are given for each]. Our 50-sample spot-check human verification checked tuned prompts to confirm that additions were genuinely irrelevant and that negations successfully inverted meaning [if feasible, expand the audit sample or report additional details about perturbation quality in the final submission].
>
>
> 2. “How frequently do scores increase under delete/negate/add actions? Do you consider these cases noise, expected behavior, or evidence of inconsistency between scoring criteria/judges?”
> In WildBench, these are rare for addition and negation across all perturbation levels: addition increases occur in only 0.1–0.5% of cases, and negation increases in 0.1–2.0%. The main exception is 25% deletion, where score increases occur in 7.0% of cases, dropping to 1.2% at 50% deletion and below 1% thereafter. These increases are almost always +1 point, consistent with boundary noise in discrete holistic scoring.
> | Degree | Addition | Deletion | Negation |
> |--------|----------|----------|----------|
> | 25% | 0.5% (5) | 7.0% (72) | 2.0% (20) |
> | 50% | 0.2% (2) | 1.2% (12) | 0.3% (3) |
> | 75% | 0.1% (1) | 0.7% (7) | 0.2% (2) |
> | 100% | 0.1% (1) | — | 0.1% (1) |
> We therefore view the low-frequency increases as judge variance rather than invalidity: they are unsystematic and do not disrupt the overall degradation trend (R² > 90% in most settings). We do acknowledge that non-monotonic cases can exist in principle, especially for light deletion, where removing a small amount of content may occasionally improve an overly verbose response, but this effect disappears at higher perturbation intensities.
>
> 3. “Can you distinguish whether these deficiencies primarily stem from the rater model or the scoring criterion design?”
> We agree the paper should more clearly distinguish end-to-end benchmark sensitivity from source-specific causal attribution. To address this, we now include two additional analyses.
> First, we conduct a rubric audit to isolate the coverage component. We used Claude Sonnet 4.6 to classify each rubric criterion/checklist item in HealthBench and WildBench according to whether it explicitly penalizes irrelevant or extraneous content. By this analysis, only 6.3% of HealthBench criteria and 3.9% of WildBench criteria do so.
>
> Second, we run a small remediation experiment on 100 samples from HealthBench and 100 from WildBench. We append three checklist items targeting padding, arbitrary inversion, and missing content, then rescore the same perturbed responses through the original judge pipeline. Clean-response scores remain essentially unchanged, indicating that the added items do not penalize unperturbed outputs. In contrast, perturbation sensitivity improves substantially, especially for addition:
>
> | Intervention | HealthBench | | WildBench | |
> |--------------|-------------|--|-----------|--|
> | | % Change in Slope (β) | Factor | % Change in Slope (β) | Factor |
> | Addition | +147% | 2.5x | +16% | 1.2x |
> | Negation | +52% | 1.5x | +2% | 1.0x |
> | Deletion | Not run | — | 0% | 1.0x |
>
> These results suggest that at least part of the observed addition insensitivity is due to rubric coverage rather than judge failure alone. At the same time, we agree that full disentanglement across rubric, judge, and aggregation is not yet complete; isolating rater-specific effects would require rescoring with alternative judges, which we leave as important future work.
>
> 5. “Potential confusion caused by length/formatting issues introduced by perturbations”
> On length and format effects specifically: our negation perturbations are applied semantically via Gemini 3 Flash, in which it is instructed to preserve sentence structure and length by design.
> For addition, injected sentences increase response length, which could in principle affect scores independently of content. We agree this limitation should be discussed more clearly, that score changes may reflect secondary effects related to length, formatting, coherence, or style. Our goal was to audit benchmark behavior under the outcome of simple perturbation procedures rather than fully decompose all mechanisms of score change (i.e. to what degree length is a factor, since it is difficult to decouple length from the inherent perturbation mechanism, whether that is addition or deletion), but the reviewer is right that this complicates interpretation. We will state this limitation more explicitly.

---

> > ### Author Rebuttal · Reviewer_wmAy · 2026-04-03
> >
> > Thank you for your reply.
> >
> > 1. Can you explain in more detail "We do acknowledge that non-monotonic cases can exist in principle, especially for light deletion, where removing a small amount of content may occasionally improve an overly verbose response, but this effect disappears at higher perturbation intensities."?
> > 2. I'd like to know if the authors conducted any preliminary investigation into the reasons for the benchmark performance under this simple perturbation procedure, such as its length.

---

> > > ### Author Response · Authors · 2026-04-08
> > >
> > > Thank you for your response.
> > > 1. Yes, what we meant is that if an existing response is verbose, with redundancy across multiple sentences conveying the same information, then it is possible that random deletion may remove only a subset of those redundant sentences while leaving the core facts intact. In such cases, the benchmark score may remain unchanged, or in some cases even slightly increase, if the response becomes less verbose without losing essential content. However, this effect has limits: it is highly unlikely that 75% of a model’s response would be redundant to the extent that deleting that much content would still avoid removing any critical information. At higher deletion intensities, one would therefore expect important content to be removed, which is consistent with the observed reduction in benchmark scores.
> > > 2. No, we did not perform a dedicated investigation into length specifically as a reason for benchmark behavior under the perturbation procedure. We agree this is an important limitation and will clarify it more explicitly in the final version as left for future work. Our current experiments evaluate the end-to-end effect of simple perturbations, so any observed score changes may reflect not only semantic changes but also secondary factors such as length or formatting.

---

### Official Review · Reviewer_6XWV · 2026-03-14

**Soundness:** 3
**Presentation:** 3
**Significance:** 3
**Originality:** 3
**Overall Recommendation:** 5
**Confidence:** 4

**Summary:**

As AI models have improved, common heuristic evaluators are no longer effective. To evaluate long-form open-ended responses, the community has, in recent years, moved away from traditional evals and toward using LLMs as judges, leading to the rise of rubric-based benchmarks.

There has been evidence that LLMs do not perform well when the inputs are perturbed in different ways. This paper seeks to explore the robustness of LLMs as judges: given that the model's answer is perturbed in different ways, can rubric-based LLMs detect and lower the score accordingly? They perturb the model responses in 3 ways: 1] adding irrelevant sentences, 2] deleting sentences, and 3] negating the meaning of sentences. They then rescore the perturbed model answer with the existing rubric-based LLM as a judge.

They apply this to two datasets: 1] Health Bench, and 2] Wildbench.They find that both benchmarks strongly punish semantic negation and are much less responsive to irrelevant additions.  The deletion results differ across benchmarks: on HealthBench, deletion yields intermediate sensitivity: stronger than addition but weaker than negation, with addition being the least penalized perturbation. On WildBench, the ordering flips: deletion is the least-penalized perturbation, requiring over half the sentences to be deleted before a 25% score drop. Across both benchmarks, the clearest vulnerability is that irrelevant content can be injected without reliably moving the score, suggesting the rubrics lack adequate coverage for penalizing extraneous information.

The overall essence of the paper is to "stress-test existing rubric-based benchmarks and determine how robust they are in simple, intuitive ways."

**Compliance With Llm Reviewing Policy:**

Affirmed.

**Final Justification:**

The authors present work on an unexplored area of meta-evaluation of rubric-based LLM judges, while the analysis is simple in its nature. It opens pathways for further exploration; they also do it for 2 datasets, Wildbench and Healthbench, where rubric-based judges are commonly utilized. While there is more work that could solidify the paper's claims, that also justifies the paper's existence.

My overall recommendation is to accept.

**Key Questions For Authors:**

- The paper mentions that the perturbations are intuitive average case perturbations. But in reality, intuitive perturbations would not occur in isolation. Did you try anywhere where something is added while something else is deleted, etc.? While I know there might be too many combinations this way, even a small subset would be helpful.
- Why should irrelevant addition always lower the score in a rubric benchmark? Did you analyze whether HealthBench or WildBench explicitly contains relevance/concision / avoid-extraneous-content criteria? This also leads to my next question.
- Modern LLMs have become increasingly strong instruction followers, and since LLM judges are simply LLMs + judge prompts, the judge may evaluate exactly what the rubric asks and nothing more. This could explain why negation is reliably caught (it directly contradicts existing criteria) while addition is tolerated (no criteria penalize it). With that in mind, did you explore updating the rubric criteria against the perturbation that you were making? Did it work accurately? Does universally saying "there might be irrelevant additions, or adversarial perturbations" help improve the quality of a judge, or saying that they should be penalized help as well? This would be really helpful in designing universal rubric principles."
- There are works like AbstentionBench [https://arxiv.org/pdf/2506.09038], Missing in Premise Bench [https://arxiv.org/abs/2504.06514], and AbsenceBench [https://arxiv.org/pdf/2506.11440] that test whether LLMs can spot problems in their input, such as missing information or unanswerable premises. In your setup, the judge LLM receives the question plus the model response as its input, so at a high level, you're also testing whether an LLM can detect perturbations in what it's reading. The key difference is that your perturbations target the response being evaluated rather than the task itself, but the underlying capability being tested, robustness to corrupted input, seems related. How do you see your work positioned relative to these benchmarks, and do you think findings from that literature could inform which perturbations rubric judges are likely to fail on? Would you care to comment?
- For HealthBench specifically, did you test harmful irrelevant additions (e.g., extra unsafe medical advice) separately from merely off-topic filler? Those are very different failure modes.

**Limitations:**

- No proposed remediation experiments. While a large-scale study would incur extra costs, which is understandable, even a few small-sample experiments would help determine what is effective or ineffective in closing the gaps they identify.
- Only 50 samples per perturbation type are human-verified (Section 4.1). For negation, especially, which requires semantic inversion rather than simple token insertion, this is insufficient. A failed negation (one that doesn't actually invert meaning) would inflate the robustness measure.
- Treating all sentences as equal ignores the fact that for datasets like Healthbench, some sentences might carry critical safety information while others are transitional. The paper never analyzes whether perturbation of high-weight rubric-relevant sentences drives most of the observed effects.
- AdvancedIF, ResearchRubrics, and FollowBench are mentioned in the introduction, but are not really evaluated on. This is not a major limitation, but would help analyze a wider variety of rubric-based benchmarks.

**Strengths And Weaknesses:**

Strengths:
- This paper is a meta-evaluation of new generation evaluation methods, a yet underexplored domain. Earlier evaluation methods were brittle in different ways, whereas this paper examines a rubric-based LLM as a judge.
- The perturbation framework is simple and easy to reason about, which makes it practical to adopt and extend to other benchmarks beyond the two tested here.
- Helpful in designing future rubric-based LLM as a judge, and the call for perturbation tests to evaluate robustness is a good practical contribution to future LLM as a judge development (Imo akin to unit testing LLM as a judge)

Weaknesses:
- The paper treats weak penalties for irrelevant additions as a critical benchmark flaw. But this assumes people would expect a score drop when off-topic sentences are injected. Without a human study confirming that expectation, "insensitive to addition" is suggestive but not definitive evidence that something is broken.
- Random sentence-level perturbation is not particularly informative.  If deletion barely moves the score, that might just mean the deleted sentences were redundant. If addition barely moves the score, it might mean the added sentences landed in places the judge naturally skips over. The paper doesn't distinguish between "the benchmark failed to catch a problem" and "there was no real problem to catch."
- The paper can't separate three possible explanations for the failures it observes: the rubric is missing criteria that would penalize the perturbation, the judge model sees the criteria but fails to apply them, or the score aggregation method washes out local failures. The authors lean toward blaming rubric coverage (Section 5), but the experimental setup doesn't isolate which of these is actually responsible.

---

> ### Author Rebuttal · Authors · 2026-03-31
>
> We thank the reviewer for the constructive feedback and for recognizing the contributions. We respond to each of the concerns below and will incorporate these clarifications in the final manuscript:
>
>
> 1. “ Did you try anywhere where something is added while something else is deleted, etc.?”
> We thank the reviewer for highlighting an imprecision in our wording: our intent was not to present each perturbation as a typical real-world error, but to contrast our approach with adversarial or worst-case attacks. We will revise our framing to describe these as simple, non-adversarial sanity-check perturbations. We intentionally tested one perturbation at a time to keep the analysis controlled and interpretable, but agree that combinations are a natural next step and may better reflect real-world failure modes. Studying them rigorously would likely require additional human validation to determine which combinations are perceived as realistic in practice. We will consider this for the camera-ready version.
>
> 2. “The paper can't separate three possible explanations for the failures it observes”
> We agree the paper should more clearly distinguish end-to-end benchmark sensitivity from source-specific causal attribution. To address this, we now include two additional analyses:
>
> First, we conduct a rubric audit to isolate the coverage component. We used Claude Sonnet 4.6 to classify each rubric criterion/checklist item in HealthBench and WildBench according to whether it explicitly penalizes irrelevant or extraneous content. By this analysis, only 6.3% of HealthBench criteria and 3.9% of WildBench criteria do so. While roughly 30% of HealthBench criteria have negative weights, they mostly target factual correctness or incompleteness in otherwise relevant content rather than extraneous padding.
>
> Second, we run a small remediation experiment on 100 samples from HealthBench and 100 from WildBench. We append three checklist items targeting padding, arbitrary inversion, and missing content, then rescore the same perturbed responses through the original judge pipeline. Clean-response scores remain essentially unchanged, indicating that the added items do not penalize unperturbed outputs. In contrast, perturbation sensitivity improves substantially, especially for addition:
>
> | Intervention | HealthBench | | WildBench | |
> |--------------|-------------|--|-----------|--|
> | | % Change in Slope (β) | Factor | % Change in Slope (β) | Factor |
> | Addition | +147% | 2.5x | +16% | 1.2x |
> | Negation | +52% | 1.5x | +2% | 1.0x |
> | Deletion | Not run | — | 0% | 1.0x |
>
> These results suggest that at least part of the observed addition insensitivity is due to rubric coverage rather than judge failure alone. At the same time, we agree that full disentanglement across rubric, judge, and aggregation is not yet complete; isolating rater-specific effects would require rescoring with alternative judges, which we leave as important future work.
>
> 3. “ How do you see your work positioned relative to these benchmarks”
> We thank the reviewer for flagging these related and relevant works. The key distinction is that those benchmarks test whether an LLM can detect problems in the input task itself versus the response being evaluated; that said, findings from AbsenceBench and Missing in Premise are informative for ours, since they suggest models are weaker at detecting missing information and may identify a missing premise without translating it into the final reported score, both of which are consistent with our deletion results.
>
> 4. “For HealthBench, did you test harmful irrelevant additions”
> We did not separate harmful off-task medical advice from benign filler in the current submission: we agree this is important, especially for medical settings, and will clarify it as future work.
>
> 5. “The paper doesn't distinguish between “the benchmark failed to catch a problem” and "there was no real problem to catch.”
> We agree this concern is most relevant for deletion, since randomly removed sentences may be redundant. We chose random perturbation as a benchmark-agnostic baseline that avoids cherry-picking, but highly-important-aware perturbation of rubric-relevant sentences is a natural and valuable extension that we plan to add in the open-source release.
>
> 6. “Only 50 samples per perturbation type are human-verified”
> We agree that the 50-sample human audit is limited, especially for negation, and we will make this limitation more explicit and, if feasible, expand the audit or report additional perturbation-quality details.
>
> 7. “Limited Benchmarks”
> We agree broader benchmark coverage would strengthen the paper; we chose HealthBench and WildBench to study two distinct rubric-based evaluation regimes across multiple permutations in depth, within our cost constraints, and will clarify this scope choice more explicitly.

---

> > ### Author Rebuttal · Reviewer_6XWV · 2026-04-04
> >
> > While most of my questions have been adequately addressed.
> >
> > Could you explain what you will do here: "We chose random perturbation as a benchmark-agnostic baseline that avoids cherry-picking, but highly important-aware perturbation of rubric-relevant sentences is a natural and valuable extension that we plan to add in the open-source release." Do you mean you will do this by the camera-ready? What would an experimental design for this look like? Do you think LLMs are good enough for deletion, or will you, the authors, manually remove relevant sentences?
> >
> > Thanks a lot for the quick experiment on [2].

---

> > > ### Author Response · Authors · 2026-04-08
> > >
> > > Thank you for the response.
> > >
> > > To clarify, we did not mean that we would add this experiment by the camera-ready deadline. Our intent was only to identify it as a natural next step for the open-source release, since it would require a more targeted perturbation pipeline (that requires human verification) than the random deletion used in the current paper.
> > >
> > > For an importance-aware deletion design, we would first use an LLM judge to score the importance of each sentence in a response with respect to the rubric/checklist items, and then human-verify a subset to check whether the LLM-based importance labels align with human judgment. Deletions would then be applied in descending order of sentence importance, allowing a direct comparison against uniform random deletion.

---

### Decision · Program_Chairs · 2026-04-30

**Decision:**

Accept (regular)

**Comment:**

This paper addresses a timely and increasingly critical problem: the reliability of rubric-based evaluation frameworks in the era of LLM-as-a-judge. As the community rapidly adopts such benchmarks for assessing open-ended and long-form generation, understanding their robustness is both urgent and under explored.  While some reviewers raised concerns regarding causal attribution, generalizability, and the scope of evaluation, the authors have responded constructively in the rebuttal, clarifying most of the concerns. Overall the paper makes a meaningful and practical contribution to ICML community, so I recommend acceptance. Please include the rebuttal contents in the final camera ready version.